# Using the Avocado to Test the Satiety Effects of a Fat-Fiber Combination in Place of Carbohydrate Energy in a Breakfast Meal in Overweight and Obese Men and Women: A Randomized Clinical Trial

**DOI:** 10.3390/nu11050952

**Published:** 2019-04-26

**Authors:** Lanjun Zhu, Yancui Huang, Indika Edirisinghe, Eunyoung Park, Britt Burton-Freeman

**Affiliations:** Center for Nutrition Research, Institute for Food Safety and Health, Illinois Institute of Technology, Chicago, IL 60616, USA; lzhu29@hawk.iit.edu (L.Z.); yancui22@hotmail.com (Y.H.); iedirisi@iit.edu (I.E.); epark4@iit.edu (E.P.)

**Keywords:** avocado, dietary fat, dietary fiber, satiety, obesity, gut peptides

## Abstract

This study aimed to investigate the satiety effects of isocalorically replacing carbohydrate energy in a meal with avocado-derived fats and fibers. In a randomized 3-arm, 6-h, crossover clinical trial, thirty-one overweight/obese adults consumed a low-fat control meal (CON, 76% carbohydrate, 14% fat as energy, 5 g fiber, ~640 kcal) or high-fat meals similar in total fat and energy, but increasing avocado-derived fat and fiber content from half (HA, 68 g; 51% carbohydrate, 40% fat as energy, 8.6 g fiber) or whole avocado (WA, 136 g; 50% carbohydrate, 43% fat as energy, 13.1 g fiber) on three separate occasions. Visual analog scales (VAS) assessed subjective satiety over 6 h. Hormones associated with satiety/appetite were measured in blood collected immediately after VAS. Stepwise multiple regression analysis was used to assess the relationship of VAS with hormones in WA and CON. Hunger suppression was enhanced after the WA compared to CON meal (*p* < 0.01). Subjects indicated feeling more satisfied after both HA and WA than CON (*p* < 0.05). Fullness was greater after CON and WA vs. HA (*p* < 0.005). PYY and GLP-1 were significantly elevated after WA vs. CON (*p* < 0.05), while insulin was significantly higher after CON vs. WA (*p* < 0.0001). Ghrelin was suppressed more by CON than WA (*p* < 0.05). Regression analysis indicated PYY was associated with subjective satiety after WA, whereas increased insulin predicted changes in subjective satiety after CON. Replacing carbohydrates in a high-carbohydrate meal with avocado-derived fat-fiber combination increased feelings of satiety mediated primarily by PYY vs. insulin. These findings may have important implications for addressing appetite management and metabolic concerns.

## 1. Introduction

The prevalence of obesity has increased dramatically over the past few decades afflicting people of all ages, races/ethnicities and both sexes [1,2,3,4,5,6]. Accordingly, obesity research efforts have focused on identifying key factors contributing to obesity development and reduction. Processes of satiation and satiety and their impact on food intake have been areas of research concentration. The macronutrient profile of a meal, including fiber, influences satiation and satiety [7,8,9,10,11]; however, the optimal composition to maximize satiety remains controversial.

Fats are considered less satiating than carbohydrate and protein; although in energy replacement and energy density controlled paradigms this is not necessarily the case [12,13,14]. Characteristic properties of fats, such as fatty acid chain length and saturation also have been studied, yielding conflicting results on satiety outcomes [15]. Similarly, inconclusive results relative to fibers’ properties on satiety have been reported [16]. Studying nutrient components out of their natural matrix may be one factor that influences satiety results, but population characteristics is an important consideration. Satiety research on fats or fibers has been conducted primarily in healthy weight adults [8,15,16] limiting extrapolation to effects in overweight and obese individuals that may experience greater benefit from certain dietary manipulations [8]. 

Fats and some fibers slow gastric emptying, delay nutrient absorption, modulate glucose and insulin responses, and alter gut hormones involved in satiety [7,17,18,19]. Fats are potent stimulators of certain satiety peptides, including cholecystokinin (CCK), peptide YY (PYY) and glucagon-like peptide (GLP)-1 [20,21,22,23]. Delaying absorption of fat with fibers could result in increased release of these peptides and enhance satiety, due to similarities in their physiological effects. Previous work suggests that under energy equivalent conditions, increasing the fat content of a meal or incorporating foods inherently containing fiber into a low-fat meal can enhance satiety over an extended post-meal time period, an effect related to the CCK response post-meal [24]. Few studies have focused on the role of nutrient combinations (particularly fat and fiber) to enhance satiety.

Avocados are a unique fruit that contains both fats and dietary fibers. One medium size, fresh Hass avocado (~136 g) is about 72% water and contains ~13.3 g monounsaturated fats, 10 g fiber, and a variety of carotenoids and other bioactive components [25]. Wien and colleagues (2013) reported that adding approximately half an avocado to a lunch meal suppressed the desire to eat and increased how satisfied participants felt over 5 h compared to a meal without avocado [26]. However, adding avocado increased the energy content of the meal. In the same study, postprandial insulin concentrations were reduced after replacing some fat and carbohydrate energy (salad dressing and cookie portions) with avocado [26]. These data suggest that strategic manipulation of meals with avocado, a source of fat and fiber, could promote both satiety and metabolic benefits. Achieving both benefits without increasing energy would be ideal, particularly for individuals with weight or glucose control concerns.

Therefore, the present study examined the satiety effects of replacing carbohydrate energy in a meal with a half or a whole avocado compared to a control meal without avocado in overweight/obese individuals. The primary research outcome was the change in subjective measures of satiety in response to meals over 6 h. Secondary outcomes were changes in variables such as tiredness and alertness, and characterization of the association between satiety/appetite related hormones (PYY, GLP-1, ghrelin and insulin) and subjective satiety after the whole avocado vs. control meals. The working hypothesis is that under energy equivalent conditions, replacing carbohydrate with fat-fiber combination from avocados will enhance satiety and the effects will be related to the meal responsive changes in satiety-mediating hormones. 

## 2. Materials and Methods 

### 2.1. Ethics and Study Design

This study was approved by the Institutional Review Board (IRB) of Illinois Institute of Technology (IIT) in Chicago, IL and registered with ClinicalTrials.gov (NCT02479048). The study was conducted according to the guidelines laid down in the Declaration of Helsinki and the International Conference on Harmonization-Good Clinical Practice (ICH-GCP). All subjects provided written informed consent. This study was part of a larger study examining the effects of avocado on metabolic endpoints, vascular reactivity and lipoprotein metabolism [27]. The study was a randomized, three-arm, single-blinded, acute, crossover design conducted from 2015–2016 at the Clinical Nutrition Research Center (CNRC) at the Illinois Institute of Technology (Illinois Tech, Chicago, IL, USA). 

### 2.2. Subjects

Thirty-nine subjects (21 men and 18 women) were enrolled and randomized in the study; 31 subjects (15 men and 16 women) completed all protocol specified procedures (Figure 1) [27]. Eligible subjects were men and women 20–65 years of age and BMI between 25 and 35 kg/m^2^ with elevated fasting glucose (5.0–6.4 mmol/L) and insulin (no greater than 90.3 pmol/L) concentrations. Participants also had to be nonsmokers and be in relatively good health with no previous history or current clinical evidence of cardiovascular, metabolic, respiratory, renal, gastrointestinal or hepatic diseases. Individuals taking medications or dietary supplements that would interfere with study outcomes, anyone reporting allergies or sensitivity to study products, and women who were pregnant or lactating were excluded. Female participants were studied during the follicular phase of their menstrual cycle. Each subject was studied once. 

### 2.3. Study Meals and Intervention

There were three breakfast test meals: one was a low fat, high carbohydrate meal (control meal, CON) and two meals were similar in energy and energy density to CON but contained either a half (HA) or a whole (WA) of a fresh, medium-sized Hass avocado (Appendix A). Meals consisted of a bagel sandwich (with or without avocado), fresh honeydew melon, oatmeal, and a lemonade-flavored drink. Nutrient and energy content for each meal appear in Table 1. Avocado-derived fat and fiber increased with increasing avocado content in the meal, with WA containing > 2/3 avocado-derived fat and fiber. HA was supplemented with butter fat to adjust total fat content between the two experimental (avocado) meals. Bagel sandwiches were used for manipulating components by adding avocado into hollowed bagels and using cream cheese and butter to manage desired fat levels and textural quality. Green leafy lettuce was used in all bagel sandwiches to help mask substitutions and control for color, visual appeal, and texture. 

All meals were prepared in the metabolic kitchen at the CNRC under the supervision of the registered dietitian following strict food safety standards. Subjects came to the laboratory on three separate occasions and consumed each meal once based on a randomly assigned sequence by computer-generated randomization allocation list. 

### 2.4. Assessments: Visual Analog Scale (VAS)

The VAS tool measured subjective satiety [28] and other variables at baseline and at seven time-specific intervals post meal [24]. The VAS questions included assessment of subjective feelings of fullness, hunger, desire to eat, prospective consumption, and how satisfied subjects felt at the moment of VAS collection. Other variables, including thirstiness, feelings of nausea, tiredness, and alertness also were assessed. Subjects were instructed to read each question carefully every time and indicate how they felt at the exact moment by drawing a straight vertical line on a 100 mm horizontal line scale anchored with “not at all” to “extremely.”

### 2.5. Assessments: Blood Collection and Analysis

Whole blood samples were collected into a 4 mL ethylenediaminetetraacetic acid (EDTA)-coated vacutainer tubes, inverted gently and immediately placed on ice. Within 30 min of collection, all blood samples were centrifuged at 453× *g* for 15 min at 4 °C. Plasma aliquots then were collected and stored at –80 °C until analysis. Prior to centrifugation, the following were added: 230 μL of aprotinin (Fisher Scientific, Fair Lawn, NJ, USA) and 40 μL of dipeptidyl peptidase IV inhibitor (EMD Millipore, Temecula, CA, USA) for GLP-1_7-36_ and PYY_3-36_ and 30 μL of 4-hydroxymercuribenzoic acid for ghrelin. Glucose and insulin were analyzed using Randox Daytona Auto Clinical Analyzer (Randox, Antrim, UK) by enzyme-based assay kits (cat. no. GL3815, Randox, Antrim, UK) and immunoturbidimetry assay (cat. no. KAI071, Kamiya Biomedicals, Tukwila, WA, USA) respectively. Enzyme-linked immunoassay kits (Phoenix Pharmaceuticals, Inc. Burlingame, CA, USA) were used to analyze total ghrelin, GLP-1_7-36_, and PYY_3-36_. Quality controls suggested by the manufacturers were used in all assays.

### 2.6. Summary of Study Procedures

Study procedures are described in detail elsewhere [27]. Briefly, subjects were asked to follow a strictly limited avocado, olive oil, nuts and polyphenolic diet for at least three days prior to the study day and throughout the study duration, while maintaining their usual eating and physical activity patterns. In addition, subjects were asked to restrict intake of alcohol, coffee, tea, and other caffeinated beverages and limit their physical activity 24 h prior to a testing day. 

Subjects were asked to consume a standardized dinner meal on the night before each study day to control variance from prior food and beverage intake. Subjects arrived at the research center in a fasted state (8–12 h fasting), well-hydrated and well-rested (+/− 1 h usual sleep pattern). Vigorous exercise, alcohol and caffeine intake were restricted 24 h prior to the visit. After reviewing subjects’ compliance and study readiness, participants completed their first set of VAS subjective satiety questions followed by placement of an intravenous catheter inserted into their antecubital vein for a fasting/baseline blood sample (0 h). Participants then were given one of the three test meals based on their assigned computer generated randomization sequence. At 0.5, 1, 2, 3, 4, 5, and 6 h after the breakfast meal, VAS again were administered. Immediately after VAS completion, blood samples were collected via the catheter. The study day visit procedures are shown in Figure 2. Each study visit occurred at least one week apart (7–10 days).

### 2.7. Statistical Analysis

Demographic characteristics of study participants were analyzed using descriptive statistics. Subjective data from VAS and biological samples were analyzed by mixed-model analysis of repeated measures using PROC MIXED [29]. All models included meal, time and meal- by-time interaction as main factors with subject as the blocking variable. VAS data were normalized to subject’s own baseline, which also corrected for baseline variance. Biological sample data were analyzed using all values with baseline values as a covariate and are presented as mean ± standard error (SE) scores over time using the Statistical Analysis System (SAS) generated least square means ± SE of the response. Area under the 6 h response curves using the trapezoidal method after controlling for baseline (iAUC) also were calculated for biological sample data. Data not conforming to expected distributional assumptions were log-transformed. For non-normally distributed data, such as ghrelin iAUC, the Wilcoxon rank sum test was performed in SAS. 

Hormone data are presented as time by concentration curves. Stepwise multiple regression analysis was performed using SPSS [30] to investigate the association of gut hormones and insulin and subjective satiety responses using only CON and WA data for each gut peptide, insulin, and VAS change from baseline. The probability of *F*-to-enter is 0.05, probability of *F*-to-remove is 0.10. A two-tailed *p*-value < 0.05 was considered significant. Power calculations based on paired t-test indicated that a sample size of thirty subjects would provide >80% power to detect differences of 25% or greater with SD of 18 in VAS or gut hormone variables between avocado-containing meals and CON meal. 

## 3. Results

### 3.1. Subjects and Test meals

Thirty-one men and women aged 38 ± 10 years with BMI 29.0 ± 2.4 kg/m^2^ completed the study. Subjects’ characteristics are shown in Table 2. All test meals were well liked (mean ± SE pleasantness score for CON, HA, WA meals: 68.9 ± 5.5, 71.2 ± 5.4, 68.5 ± 6.0, respectively on a scale of 0–100, *p* > 0.05). No adverse events related to test meals were reported.

### 3.2. Subjective Satiety

All meals significantly suppressed hunger, desire to eat and prospective consumption and increased fullness and satisfaction (time effect *p* < 0.05 for all) (Figure 3a–e). Meal-specific effects were apparent for changes in hunger, fullness and how satisfied subjects felt over 6 h (*p* < 0.05) and marginally for prospective consumption (*p* = 0.11). In general, hunger was significantly suppressed over time by WA compared to CON (Figure 3a), whereas changes in ratings of fullness were not different between WA and CON but were different from HA (Figure 3b). Changes in desire to eat were not different among meals (Figure 3c). Meal related changes in how much subjects wanted to eat (prospective consumption) over 6 h suggested greater suppression with increasing amounts of avocado in the meals (*p* = 0.11, Figure 3d). How satisfied subjects felt were enhanced after both avocado meals compared to CON (Figure 3e, *p* < 0.0005 for both). Mean scores for thirst, nausea, and alertness did not differ among meals, whereas participants reported being less tired (*p* < 0.05) after WA (29.7 ± 2.4) than from HA (33.7 ± 2.3).

### 3.3. Gut Peptide and Insulin Responses

Total ghrelin was significantly suppressed starting at 1 h after the CON and WA meal reaching a nadir at 2 h and returning to baseline at 6 h (Figure 4a). Between meal analysis indicated that CON produced a significantly greater suppression of total ghrelin compared to WA (meal effect, *p* < 0.05; meal × time interaction, *p* < 0.05), which was supported by the iAUC analysis indicating a 35% greater suppression of total ghrelin after the CON vs. WA meal (*p* = 0.01, Table 3). GLP-1 increased significantly from baseline after both CON and WA meals (*p* < 0.05) and returned to baseline by 4 h (Figure 4b); however, the pattern of response differed, in that, GLP-1 reached peak concentrations at ~30 min after WA and at ~1 h after CON. Incremental AUC indicated WA induced ~77% greater exposure to GLP-1 compared to CON over 6 h. Consistent with the GLP-1 response pattern, PYY was significantly elevated after WA meal compared to CON (~2.5-fold concentrations, *p* < 0.001). Further, PYY peaked ~30 min earlier after WA meal vs. CON followed by a slower rate of clearance (Figure 4c). Insulin concentrations after all meals were reported previously [27]. Briefly, insulin increased after all meals and returned to baseline by 4 h post-meal; however, insulin concentrations were significantly reduced after the avocado-containing meals compared to CON. Insulin iAUC after the WA meal was reduced by 31% compared to the CON meal (*p* < 0.0001, Table 3). 

### 3.4. Relationship of Hormones and Subjective Satiety

All hormones played a role in the subjective satiety responses to meals (Table 4); however, the strength of the relationship depended on which meal was ingested (CON or WA) and the variable investigated. Overall, PYY predicted 20–30% of the changes in hunger, fullness, desire to eat and prospective consumption after WA, but only 3–5% of the variance in satiety response after the CON meal. In contrast, insulin predicted subjective satiety responses after CON ranging from 23–32% depending on the subjective satiety variable, whereas insulin was not significant in the regression models with WA. GLP-1 explained much of the fullness response for CON. Ghrelin had a relatively minor role in subjective responses for WA. 

## 4. Discussion

The purpose of this study was to investigate the satiety effects of avocados, a unique food source containing both fats and fibers, in an isocaloric replacement paradigm, substituting carbohydrate energy in a low-fat, high carbohydrate (Control, CON) meal with half or whole avocado. The potential impact of the fat-fiber combination derived partly or predominantly from avocados, independent of fat content, was assessed by adjusting total fat in the avocado meals. Study results indicated that replacing carbohydrate with fats and fibers derived from avocados without increasing energy or energy-density enhanced the satiety value of meals in overweight and obese individuals as evidenced by greater hunger suppression after WA compared to CON (WA > CON = HA) and increased feelings of how satisfied subjects felt over 6 h (WA = HA > CON). Satiety induced by WA was associated mostly with PYY responses. In contrast, satiety induced by CON was related mostly to insulin responses. These findings suggest isocaloric dietary manipulation with a whole avocado promotes favorable metabolic responses in addition to enhancing satiety and reducing motivation to eat. 

Optimal meal composition to maximize satiety remains controversial. In general, protein is considered to be the most satiating followed by carbohydrate and then fat [31]. Importantly however is whether energy and volume is controlled in study designs comparing macronutrients and satiety. Fat has a clear disadvantage because it has twice the energy density of other macronutrients. Early evidence suggests that when palatability and energy density is held constant, subjective satiety, and energy intake as a measure of satiating effect was not different between carbohydrate and fat preloads [13,32,33]. This has been observed when foods were ingested orally or infused intravenously or intragastrically [12]. Likewise, texture, and other sensory cues must be considered when investigating satiety effects of macronutrients. In the present study, energy, energy density, palatability, and various sensory factors (including visual appeal) were matched closely. All meals looked and had texturally similar components. All meals were well liked (pleasantness mean score >60 for all meals and not significantly different) and all suppressed hunger and induced satiety. However, compared to CON, the WA meal produced a significantly greater reduction in hunger and increase in how satisfied participants felt after the WA meal compared to CON. Subjects also reported feeling more satisfied after the HA meal compared to the CON, while other subjective satiety responses were not significantly suppressed/enhanced by HA. The HA meal had similar total fat content to the WA meal, but less avocado-derived fat and fiber than WA meal. These data suggest that having a higher level of fat in the meal is associated with subjects feeling more satisfied, whereas increasing both the avocado-derived fat and fiber combination offers additional satiety-inducing benefits influencing several subjective variables. Whether it is the fat component or the fiber component driving the effect is the subject for subsequent research. Findings from this study suggest that replacing carbohydrate energy with fat energy in combination with a source of inherent (rather than added) fiber, as demonstrated here with avocados, can increase the satiety value of a meal compared to a low-fat high-carbohydrate meal. 

The control of food intake, satiety and appetite involves complex processing of non-homeostatic and homeostatic factors, the latter including physiological status and the dynamics of fuel metabolism and episodic gut hormones in response to food intake. Flint et al. reported that insulin concentrations were inversely correlated with feelings of hunger, whereas glucose concentrations were not related to hunger or satiety [34]. In this current study, feelings of hunger were significantly suppressed with the WA meal, yet insulin was also markedly reduced [27]. GLP-1 and PYY are anorexigenic hormones and produce feelings of satiety and inhibit food intake [35,36]. Fat is a potent stimulator of GLP-1 and PYY, explaining their significant elevation after WA meal compared to CON meal. PYY was identified as the primary contributor to the satiety response of WA, with GLP-1 and ghrelin contributing only modestly. In contrast, ghrelin suppression was greater after CON compared to the WA meal, consistent with carbohydrate being more potent than fat or protein to suppress ghrelin [37,38]. However, analysis did not reveal ghrelin as playing a significant role in hunger or other subjective satiety responses to CON. Instead, insulin was consistently identified as explaining much of CON meal-associated satiety variance, except in the case of fullness, in which GLP-1 contributed 32% and insulin added an extra 8%. These findings suggest that ***how*** satiety is achieved through biological signaling may have important implications. Achieving satiety through sufficient release of pre-absorptive satiety peptides may be ideal versus exaggerated insulin responses to meals.

Insulin is a master hormone in fuel trafficking and metabolism and has varied roles centrally, including food intake regulation. A dietary pattern that is associated with elevated insulin has been identified with increasing risk of hyperglycemia, metabolic syndrome and insulin resistance [39,40,41], particularly in an overweight/obese population. Hyperinsulinemia and insulin resistance in the periphery can produce hypoinsulinemia and/or modified/dampened insulin signaling in the brain [42,43,44]. Individuals in the current study were overweight or obese and had elevated fasting insulin concentrations with insulin resistance [27]. Findings suggest that replacement strategies for carbohydrate with avocado fats and fibers reduced the post meal insulin demand, increased satiety gut hormone concentrations and improved the subjective profile of satiety. Whether the effects are due to improved insulin sensitivity peripherally or centrally to augment satiety signaling cannot be determined in the present research, but provides the basis for follow up mechanistic work in appropriate models. From a practical standpoint, the population studied represents a typical cohort of middle-aged people at risk for cardio-metabolic disease—a point when realistic dietary changes can make a significant impact on reversing the disease risk trajectory.

This study has strengths and limitations. An important strength was the application to real life. Isocaloric replacement of carbohydrate for fat is not easy yet may have advantages for appetite and post-meal metabolic control. Strategic manipulation of menu items maintaining consistent energy density and sensory variables was achieved using readily available foods in grocery stores. Avocados provided healthy fat and fibers, adding to the simplicity of the macronutrient exchanges. One study limitation was the lack of a test meal to measure subsequent food/energy intake 6 h after meals as the meal-related time course of subjective satiety (particularly later in the postprandial period) suggested an effect on subsequent meal intake may have been plausible. Not separating the specific fat type and fiber effect also may be a limitation. Finally, gut hormones were not measured after HA. This information may have provided insight on the observed differences in fullness ratings between HA meal and CON and WA meals. 

## 5. Conclusions

The increasing rates of obesity over the last several decades with limited success in reversing trends has intensified research to gain a better understanding of how foods and ingredients impact appetite and satiety, energy intake, and body weight management. For years, fat has been targeted as one of the main causes of obesity, in part due to behaviors associated with passive over-consumption. However, carbohydrates have now come under scrutiny in association with obesity. It is timely to re-examine the issue of specific macronutrients in the context of their interactions in the diet and understanding ***how*** satiety and food intake control is achieved in different populations. The implications of metabolically healthy responses as well as managing subjective appetite warrant further research with avocados and other fat-fiber combinations, particularly in overweight and obese populations with metabolic concerns. 

## Figures and Tables

**Figure 1 nutrients-11-00952-f001:**
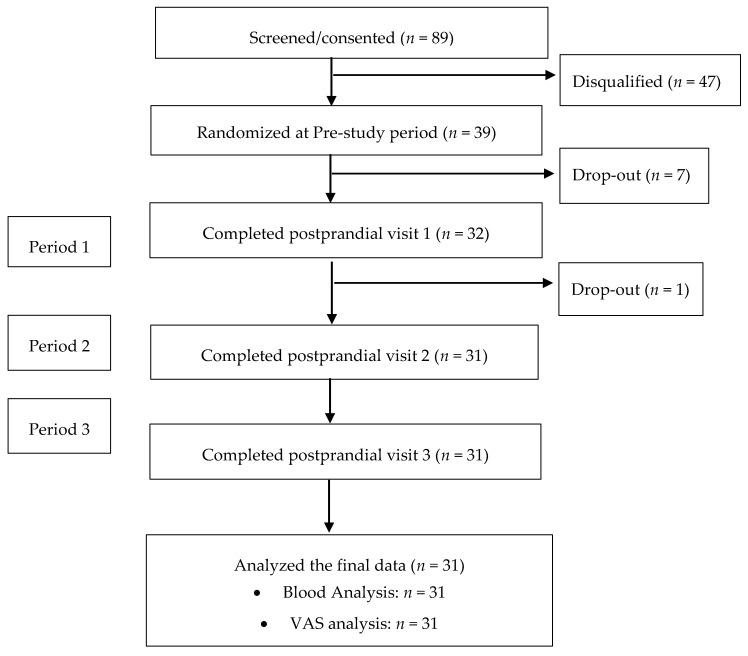
Consolidated Standards of Reporting Trials (CONSORT) flow diagram of the study. VAS, visual analog scale.

**Figure 2 nutrients-11-00952-f002:**
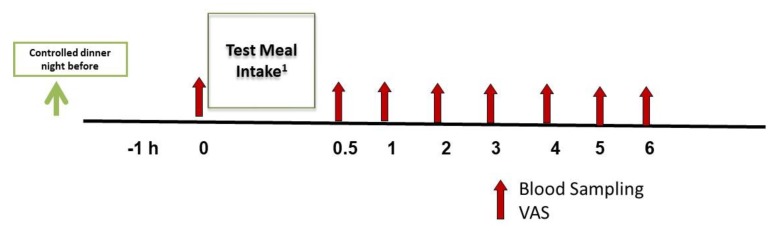
Six hour postprandial study day schema. VAS, visual analog scales. ^1^ One of three test meals provided at each study day visit. Test meal sequence randomly assigned. Each subject consumes each test meal one time.

**Figure 3 nutrients-11-00952-f003:**
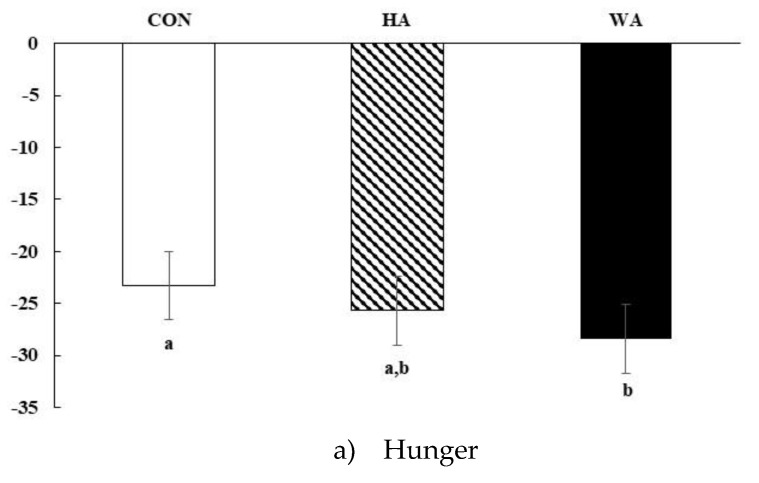
Mean changes in Visual Analogue Scale (VAS) scores as a measure of subjective satiety after meals: (**a**) hunger, (**b**) fullness, (**c**) desire to eat, (**d**) prospective consumption, (**e**) satisfaction after consumption of CON, Control meal; HA, Half Hass avocado meal; or WA, Whole Hass avocado meal. Main effects of meal *p* < 0.05 indicated by different letters, time *p* < 0.0001, and meal × time *p* NS. *N* = 31.

**Figure 4 nutrients-11-00952-f004:**
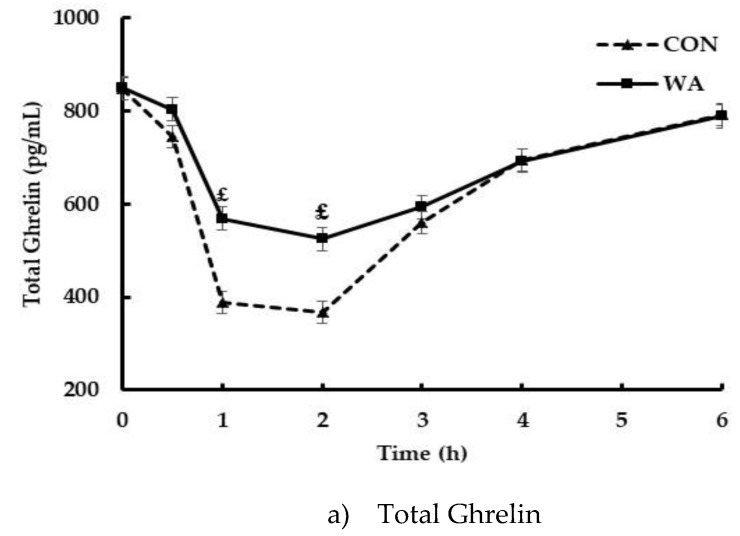
Postprandial profiles of: (**a**) total ghrelin, (**b**) Glucagon-like peptide, GLP-1_7-36_, (**c**) peptide YY, PYY_3-36_ after consumption of avocado or control meals (WA or CON, respectively). Data were analyzed by PROC MIXED using SAS 9.4. Main effects meal *p* < 0.01, time *p* < 0.01, and meal by time *p* < 0.01 were observed. ^#, €,^
^£^ Level of significance between treatments at each time point, ^#^
*p* < 0.05, ^€^
*p* < 0.01, ^£^
*p* < 0.0005. Data are means ± SE, *n* = 31. Abbreviations: CON, Control meal; HA, Half Hass avocado meal; or WA, Whole Hass avocado meal.

**Table 1 nutrients-11-00952-t001:** Nutrient composition of breakfast test meals for postprandial study day visits ^1^.

Nutrient	Control Meal CON	Half Avocado Meal HA	Whole Avocado Meal WA
Calories (kcal)	637	618	642
% Carbohydrate	75.7	50.8	50.3
% Fat	14.0	39.8	43.0
% Protein	11.5	11.8	10.4
Dietary Fiber (g) ^2^	4.9	8.6	13.1
Sugar (g)	59.4	26.5	25.5
Total Fat (g) ^3^	9.9	27.3	30.7
Saturated Fat (g)	4.9	11.3	8.6
Monounsaturated Fat (g)	2.1	10.8	15.8
Polyunsaturated Fat (g)	1.2	2.6	3.6
Energy Density (kcal/g)	2.1	2.0	2.1

^1^ Nutrients of food ingredients analyzed by Food Processor Pro SQL Edition by ESHA (Version 10.15.41 ESHA Research, Salem, OR, USA). ^2^ Grams, g (and% of total) of dietary fiber from avocado in HA and WA meals: 4.6 g (53%) and 9.1 g (70%), respectively. ^3^ Grams, g (and% of total) of fat from avocado in HA and WA meals: 10 g (37%) and 19.9 (65%), respectively.

**Table 2 nutrients-11-00952-t002:** Demographic information (Mean ± SD) ^1^.

Characteristic		Total Subjects (*n* = 31)
**Age (years)**		38 ± 10
**BMI (kg m^−2^)**		29 ± 2
**Mid-point waist circumference (cm)**		93 ± 10
**Race/Ethnicity, *n***	Cau:AA:AS:His	9:13:5:4
**Gender, *n***	Male:Female	15:16

^1^ Abbreviations: AA, African America; AS, Asian; BMI, Body Mass Index; Cau, Caucasian; cm, centimeter; His, Hispanic; kg, kilogram; m, meter; mg dL^−1^, milligram per deciliter. Data obtained from screening visit and presented as mean ± SD for *n* = 31.

**Table 3 nutrients-11-00952-t003:** Gut peptide and insulin hormone iAUC over 6-hour postprandial day ^1^.

Hormones	WA	CON	*p* Value
Ghrelin iAUC (pg min·mL^−1^)	−1000.7 ± 147.3	−1345.7 ± 142.4	0.01 *
GLP iAUC (pg min·mL^−1^)	130.4 ± 12.3	73.5 ± 11.9	0.002
PYY iAUC (pg min·mL^−1^)	239.3 ± 19.5	73.8 ± 18.8	<0.0001
Insulin iAUC (pmol·L^−1^)	4285.5 ± 562.8	6221.3 ± 557.9	<0.0001

^1^ Values represent mean ± standard error of the mean of the calculated incremental area under the curve (iAUC). * Ghrelin iAUC: the Wilcoxon rank sum test was performed in SAS. Abbreviations: CON, Control meal; WA, Whole Hass avocado meal.

**Table 4 nutrients-11-00952-t004:** Association between subjective satiety ratings and satiety hormones in overweight/obese participants ^1^.

CON						WA					
Variable	^2^ Beta	*t*	*p*	*R* ^2^	Cumulative *R*^2^	Variable	^2^ Beta	*t*	*p*	*R* ^2^	Cumulative *R*^2^
**Hunger**						**Hunger**					
Insulin	−0.33	−4.40	0.000	0.22	0.22	PYY	−0.36	−3.74	0.000	0.30	0.30
PYY	−0.22	−2.94	0.004	0.03	0.25	GLP	−0.29	−2.92	0.004	0.02	0.32
GLP	−0.00	−0.04	0.966			Ghrelin	−0.17	−2.91	0.004	0.03	0.35
Ghrelin	−0.11	−1.61	0.109			Insulin	−0.01	−0.09	0.928		
**Fullness**						**Fullness**					
GLP	0.39	6.17	0.000	0.32	0.32	PYY	0.44	6.96	0.000	0.20	0.20
Insulin	0.34	5.38	0.000	0.08	0.40	Insulin	0.11	1.03	0.305		
PYY	0.12	1.65	0.100			GLP	0.06	0.55	0.581		
Ghrelin	0.06	1.06	0.292			Ghrelin	0.03	0.47	0.640		
**Desire to Eat**						**Desire to Eat**					
Insulin	−0.31	−4.28	0.000	0.23	0.23	PYY	−0.53	−8.45	0.000	0.24	0.24
PYY	−0.29	−4.01	0.000	0.05	0.28	Ghrelin	−0.16	−2.54	0.012	0.02	0.27
GLP	−0.01	−0.12	0.907			Insulin	−0.09	−0.93	0.352		
Ghrelin	−0.32	−0.49	0.623			GLP	−0.17	−1.68	0.095		
**Prospective Consumption**						**Prospective Consumption**					
Insulin	−0.34	−4.46	0.000	0.29	0.29	PYY	−0.52	−8.28	0.000	0.23	0.23
PYY	−0.18	−2.32	0.021	0.04	0.33	Ghrelin	−0.16	−2.55	0.012	0.02	0.26
GLP	−0.16	−2.20	0.029	0.02	0.34	Insulin	−0.09	−0.92	0.359		
Ghrelin	−0.08	−1.26	0.210			GLP	−0.16	−1.51	0.132		

^1^ Stepwise criteria: Probability-of-F-to-enter ≤ 0.05, Probability-of-F-to-remove ≥ 0.10. Gut hormone (GLP-1, PYY and Ghrelin) and insulin concentrations were performed as the independent variable and magnitude of change from baseline visual analog scale (VAS) scores (hunger, fullness, desire to eat, and prospective consumption) were performed as dependent variables. *n* = 29. ^2^ Standardized coefficient. Abbreviations: CON, Control meal; GLP-1, Glucagon-like peptide-17-36; PYY, Peptide YY3-36; WA, Whole Hass avocado meal; *R*^2^, *R* squared.

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
