# Peer review of "Using the Avocado to Test the Satiety Effects of a Fat-Fiber Combination in Place of Carbohydrate Energy in a Breakfast Meal in Overweight and Obese Men and Women: A Randomized Clinical Trial"

_nutrients, 2019, doi:10.3390/nu11050952_

Round 1
Reviewer 1 Report
The paper „Using the avocado to test the satiety effects of a fat-fiber combination in place of carbohydrate energy in a breakfast meal in overweight and obese men and women: a randomized clinical trial“ is a well written paper, with well explained results. It also deals with interesting subject: how to increase a satiety after a meal. The idea was to replace carbohydrates in the meal with avocado rich in fat and dietary fibers. Authors discussed about the results and they also mentioned the limitation of the study.
My recommendation is to publish the paper.
Author Response
We thank the reviewer for the constructive feedback including the uniqueness of the study.

Reviewer 2 Report
The reviewer thanks the authors for this interesting article, which analysed and reported the impact of fat content on the subjective and objective satiety change following a meal. I would like to give the following comments:
Line 16: Please specify if the fibre contribute to the total energy of carbohydrate.
Line 18: Please specify the en% of the macronutrients of the half avocado and whole avocado meal.
Line 42: References 12-14 seem to be quite old references. Would you provide more recent references on the association between fat and satiety in the energy replacement cases?
Line 57: Please check Giezenaar (2018) publication on Nutrients "Acute Effect of Substitution, and Addition, of carbohydrates and Fat to Protein on Gastric Emptying, Blood Glucose, Gut Hormones, Appetite, and Energy Intake". This may contribute to the "few studies have focused on the role of nutrient combinations to enhance satiety".
Line 87: I think the authors may not need to say "within-subject" since it is a cross-over randomised trial of repeated measures. However, I would suggest to add "acute" instead of "6h".
Line 119: Please provide a nutritional information table of the Hass avocado as supplementary file.
Line 134: Table 1. Please specify the dietary fibre (g) and fat profiles (g) from avocado for each treatment meal? Is the sugar mainly form the lemonade drink? Are there any added sugar to the control meal?
Line 160: Please specify the wash-out period.
Line 161: Is the standardised dinner meal controlled for carbohydrate energy contribution to prevent glycogen depletion after overnight fasting?
Line 178: Why not analyse incremental trajectory or incremental area under the curve?
Line 194: How many women/men were involved in the study? According Giezenaar et al. (2018) , "Effect of gender on the acute effects of whey protein ingestion on energy intake, appetite, gastric emptying and gut hormone responses in healthy young adults", there was a delay in gastric emptying time, and slower and delayed increase in plasma glucagon, CCK and GLP-1 in women than men (P<0.05). Has the gender differences been considered in the study?
Line 215: Please delete "Figure 2". Are the Figure 2 a-e the incremental area under the curve? Is there a statistically significant difference in Hunger between control meal and WA?
Line 235: The authors mentioned "elevated ghrelin at baseline". Which time point was it compared to?
Figure 3 a-c: It seems control and WA groups had the same ghrelin, GLP, PYY at the baseline. Is that correct? Is there a missing postprandial insulin response figure? Would you please also add HA treatment to all the figures?
Figure 314: Is there a dose-responsive relationship?
Author Response
The reviewer thanks the authors for this interesting article, which analysed and reported the impact of fat content on the subjective and objective satiety change following a meal. I would like to give the following comments:
è We thank the reviewer for the constructive feedback including comments below.
Line 16: Please specify if the fibre contribute to the total energy of carbohydrate.
è The compositional analysis of the meals was done using the ESHA program version (Version 10.15.41 ESHA Research, Salem OR). Dietary fibers are carbohydrates that are not digestible by mammalian enzymes, but can yield some energy from microbial fermentation to SCFA (if they are fermentable). Up to 2 kcal/g has been suggested for fibrous foods that contain on average 70% of fiber that is fermentable. Avocado fiber is suggested to be 40% soluble but it is unclear how much of the fiber is fermentable at this time. No energy has been ascribed to the fiber components in the present study by ESHA.
Line 18: Please specify the en% of the macronutrients of the half avocado and whole avocado meal.
è We added the requested information and clarified total fat content.
Line 42: References 12-14 seem to be quite old references. Would you provide more recent references on the association between fat and satiety in the energy replacement cases?
We have reviewed several recent original research studies in humans. Unfortunately, to our amazement, many studies do not control for energy when manipulating fat/carbohydrate ratios or manage portion size differences. A recent review by Carreiro et al. Ann Rev Nutr. 2016 July 17; 36: 73–103 suggest that disparity in macronutrient focused research is likely due at least in part to these issues. An important strength of the current research is that we controlled for energy, energy density, portions, and many sensory factors, which has not been done in recent fat/carbo replacement research.
Line 57: Please check Giezenaar (2018) publication on Nutrients "Acute Effect of Substitution, and Addition, of carbohydrates and Fat to Protein on Gastric Emptying, Blood Glucose, Gut Hormones, Appetite, and Energy Intake". This may contribute to the "few studies have focused on the role of nutrient combinations to enhance satiety".
è Thank you for referring us to this publication. Taken from the paper, the authors “hypothesized that the equi-energetic replacement of protein by carbohydrate and fat would result in relatively faster gastric emptying, whereas the addition of carbohydrate and fat (and hence energy) to protein would be associated with slower gastric emptying and increased suppression of ghrelin and stimulation of CCK and GLP-1, compared to control.” In review of the prepared drinks, the authors indicate the drinks were NOT matched for volume:
“Drinks were matched for taste, served in a covered cup, and prepared by dissolving whey-protein
isolate (Fonterra Co-Operative Group Ltd., Palmerston North, New Zealand) and dextrose (glucose),
and homogenizing olive oil (Bertolli Australia Pty Ltd., Unilever Australasia, Sydney, NSW, Australia)
in varying volumes of demineralized water and diet lime cordial (Bickford’s Australia Pty Ltd., Salisbury South, SA, Australia). It is also very difficult to understand how the authors might have matched taste closely. Having worked with protein powders, whey protein (WPI) has a distinct flavor and masking taste and flavor differences of a 70g load of WPI vs 14 g WPI would be extremely difficult. Nonetheless, we could reference this paper as a paper that has studied combinations.
Line 87: I think the authors may not need to say "within-subject" since it is a cross-over randomised trial of repeated measures. However, I would suggest to add "acute" instead of "6h".
è Thank you for bringing this to our attention. We have made the edit in the revised manuscript.
Line 119: Please provide a nutritional information table of the Hass avocado as supplementary file.
We have included a table showing the nutritional composition of the Hass avocado from the Hass Avocado Nutrient database (Source: https:///loveonetoday.com/nutrition/avocado-nutrition-facts-label, accessed April 22, 2019)
Line 134: Table 1. Please specify the dietary fibre (g) and fat profiles (g) from avocado for each treatment meal? Is the sugar mainly form the lemonade drink? Are there any added sugar to the control meal?
è The information has been added to the Table footnote. Great suggestion.
The sugar is mainly in the lemonade drink and brown sugar was added to the Control meal into the maple and brown sugar flavored oatmeal. The details are published in the parent Nutrients paper ref #27.
Line 160: Please specify the wash-out period.
è The washout period between visits was 7-10 days. The detail is added to paper.
Line 161: Is the standardised dinner meal controlled for carbohydrate energy contribution to prevent glycogen depletion after overnight fasting?
è Subjects had similar dinner meals between visits. Standardization was within subjects; however, all subjects were also counseled to eat extra bread (2 slices) in preparation for their fast. The general advice to prevent glycogen depletion is to have/ensure participants consume at least 150 g carbohydrate prior to an OGTT (we did not perform an OGTT, but subjects did have a “meal” test). For person following a 2000 kcal diet, this is about 30% of kcal from carbohydrate, which is much lower than our subjects’ average % carbohydrate intake.
Line 178: Why not analyse incremental trajectory or incremental area under the curve?
è We are not entirely clear what is meant by incremental trajectory, but we did perform the iAUC. We have provided this information in a new table (included as Table 3). We used line graphs to show the changing biological responses over time.
Line 194: How many women/men were involved in the study? According Giezenaar et al. (2018), "Effect of gender on the acute effects of whey protein ingestion on energy intake, appetite, gastric emptying and gut hormone responses in healthy young adults", there was a delay in gastric emptying time, and slower and delayed increase in plasma glucagon, CCK and GLP-1 in women than men (P<0.05). Has the gender differences been considered in the study?
è 15 men and 16 women were participated in the study (as shown in Table 2). We analyzed for a sex difference and it was not significant. We have detected sex differences in other studies investigating CCK and satiety after fat intake, but we did not in this study.
Line 215: Please delete "Figure 2". Are the Figure 2 a-e the incremental area under the curve? Is there a statistically significant difference in Hunger between control meal and WA?
è Figure2 a-e are the MIXED model results of the VAS rating. We are not sure why we would delete these graphs. They represent the overall subjective satiety responses to the meals based on their responses to specific questions of hunger, fullness, desire to eat, etc. These are standard output.
Line 235: The authors mentioned "elevated ghrelin at baseline". Which time point was it compared to?
è Ghrelin in an appetitive hormone. In contrast to satiety peptide hormones, it is elevated at baseline (in fasting state). Since this may be confusing, we deleted the words “elevated at baseline”
Figure 3 a-c: It seems control and WA groups had the same ghrelin, GLP, PYY at the baseline. Is that correct?
è Yes this is correct. This was an acute intervention and not a chronic feeding study. We would not expect their baselines to change from week to week.
Is there a missing postprandial insulin response figure? Would you please also add HA treatment to all the figures?
è The insulin was not previously included in the submission because it was published as part of the cardio-metabolic responses paper (ref #27). However, we have included the iAUC of WA and CON in the new Table 3. There are no data for the gut peptides in the HA meal unfortunately. This is a limitation we included in the discussion section already. The data may have helped interpret the fullness data. This was also noted in the methods section.
Figure 314: Is there a dose-responsive relationship?
è We were not able to assess the dose-response relationship of gut peptide hormones in the study. However, there was no dose-response in glucose or insulin as published in ref #27.
